# Comment on Mankowska et al. Critical Flicker Fusion Frequency: A Narrative Review. *Medicina* 2021, *57*, 1096

**DOI:** 10.3390/medicina58060739

**Published:** 2022-05-30

**Authors:** Xavier C. E. Vrijdag, Hanna van Waart, Jamie W. Sleigh, Simon J. Mitchell

**Affiliations:** 1Department of Anaesthesiology, School of Medicine, University of Auckland, Private Bag 92019, Auckland 1142, New Zealand; hanna.van.waart@auckland.ac.nz (H.v.W.); jamie.sleigh@waikatodhb.health.nz (J.W.S.); sj.mitchell@auckland.ac.nz (S.J.M.); 2Department of Anaesthesia, Waikato Hospital, Hamilton 3240, New Zealand; 3Department of Anaesthesia, Auckland City Hospital, Auckland 1023, New Zealand

We have read with great interest the review by Mankowska et al. and congratulate the authors on providing an overview of the use of critical flicker fusion frequency (CFFF) in medicine and particularly in diving medicine [1]. In their diving and hyperbaric medicine section they discuss CFFF as a measure of cognitive performance, and how this could be influenced by hyperbaric oxygen, nitrogen narcosis, and high-pressure neurologic syndrome. We would like to comment specifically on the diving medicine section of the article and point out some missing literature that provides important context for the use of CFFF in measuring nitrogen narcosis.

First, the paragraph describing nitrogen narcosis mentions the penetration of nitrogen into the lipids of neurons acting to interfere with signal transmission. This is an outdated view on the cellular mechanism of narcosis. Research on anaesthetic gases suggest an effect on ligand-gated ion-channels in the postsynaptic membrane of excitable neurones [2]. Specifically, the GABA_A_-receptor is known for binding sedative anaesthetics with consequent opening of the ionophore for chloride-ions, causing hyperpolarization of the cell membrane and thereby signal inhibition [2]. Nitrogen is also known to bind to this GABA_A_-receptor [3]. This indicates that nitrogen narcosis is not a phenomenon of a gas-lipid reaction in the bilayer but is more likely to be a gas-protein reaction within the receptors in the synapses [4].

Second, an incomplete description is given of the effects of air breathing at different diving depths on CFFF. The authors describe several studies that found a reduction in CFFF in divers breathing air at 405 kPa (the equivalent of 30 m of seawater (msw)) either inside a hyperbaric chamber or underwater [5,6,7]. This reduction in CFFF was interpreted as a reduction in cognitive performance due to nitrogen narcosis. Accordingly, it would be expected that CFFF would be further reduced when diving to 608 kPa (50 msw). However, three uncited studies where divers breathed air at 608 kPa (50 msw) inside a hyperbaric chamber or underwater did not show a further reduction in CFFF as one would expect. They found either no change [8] or an increase in CFFF [9,10]. This would indicate that there are possibly other factors influencing the CFFF measurement at 608 kPa, which casts considerable doubt on the suitability of CFFF to measure nitrogen narcosis across the plausible range of air diving exposures. A more elaborative overview of the diving CFFF literature is given elsewhere [8].

Third, in the section about “CFFF and its connection with brainwaves,” there seems to be a semantic discrepancy between CFFF, defined as ‘critical flicker fusion frequency’ and ‘flickering light.’ The description of the influence of flickering light on the electroencephalogram (EEG) has little or nothing to do with CFFF. It therefore seems out of place in a narrative review about CFFF.

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
