# Peer review of "Comment on Mankowska et al. Critical Flicker Fusion Frequency: A Narrative Review. Medicina 2021, 57, 1096"

_medicina, 2022, doi:10.3390/medicina58060739_

Round 1
Reviewer 1 Report
The authors correctly highlight that current thinking on the mechanism(s) of nitrogen narcosis has moved beyond mere lipid-solubility, to encompass a range of possibilities, though reference 2 concerns a review of the association between GABAA receptors and various clinically-relevant anaesthetic agents and I am not convinced the relationship between this review and nitrogen narcosis is strong enough to warrant including this reference. The authors then state that nitrogen is known to bind to the GABAA receptor and refer to a 2003 paper that showed pretreatment with a GABAA receptor antagonist increased the nitrogen threshold pressure in mice, (for loss-of-righting-reflex), though loss of righting reflex in mice may not be due exclusively to narcosis. On balance though, given the known association between nitrogen pressure and body sway in humans, that a GABAA antagonist has been shown associated with the ability of mice to right themselves under pressure does persuasively suggest the potential involvement of GABAA receptors. Therefore, in this first point I politely suggest the authors consider the weight of reference 2 in support of their main argument, and if they decide to steadfastly include it then that would be fine with me.
In the second point, the effect of the absolute pressure of nitrogen narcosis may be confounded by the effect of the partial pressure (or fraction) of oxygen breathed, meaning it may not be surprising that some studies report conflicting results. The authors correctly point out that a detailed review of CFFF had previously been published elsewhere (eight months before this review was submitted).
Author Response
The authors correctly highlight that current thinking on the mechanism(s) of nitrogen narcosis has moved beyond mere lipid-solubility, to encompass a range of possibilities, though reference 2 concerns a review of the association between GABAA receptors and various clinically-relevant anaesthetic agents and I am not convinced the relationship between this review and nitrogen narcosis is strong enough to warrant including this reference. The authors then state that nitrogen is known to bind to the GABAA receptor and refer to a 2003 paper that showed pre-treatment with a GABAA receptor antagonist increased the nitrogen threshold pressure in mice, (for loss-of-righting-reflex), though loss of righting reflex in mice may not be due exclusively to narcosis. On balance though, given the known association between nitrogen pressure and body sway in humans, that a GABAA antagonist has been shown associated with the ability of mice to right themselves under pressure does persuasively suggest the potential involvement of GABAA receptors. Therefore, in this first point I politely suggest the authors consider the weight of reference 2 in support of their main argument, and if they decide to steadfastly include it then that would be fine with me.
Thank you for reviewing this commentary. We have replaced reference 2 with a reference to Smith, C.R.; Spiess, B.D. The Two Faces of Eve: Gaseous Anaesthesia and Inert Gas Narcosis. Diving Hyperb. Med. 2010, 40, 68–77. As it is indeed more appropriate for the section in the manuscript.
In the second point, the effect of the absolute pressure of nitrogen narcosis may be confounded by the effect of the partial pressure (or fraction) of oxygen breathed, meaning it may not be surprising that some studies report conflicting results. The authors correctly point out that a detailed review of CFFF had previously been published elsewhere (eight months before this review was submitted).
Thank you.
Reviewer 2 Report
This is a very subtle commentary on the cFFF-review by Mankowska et al (2021).
This reviewer suggests to more clearly differentiate between the gas-lipid reaction and the gas-protein reaction and the related theories. May be further support the latter theory with a review article by Rostain et al (2011).
Author Response
This is a very subtle commentary on the CFFF-review by Mankowska et al (2021).
Thank you for reviewing our manuscript.
This reviewer suggests to more clearly differentiate between the gas-lipid reaction and the gas-protein reaction and the related theories. May be further support the latter theory with a review article by Rostain et al (2011).
Thank you pointing out to clearly articulate this difference with the appropriate reference. This has now been included in the manuscript as: “This indicates that nitrogen narcosis is not phenomenon of a gas-lipid reaction in the bilayer but is more likely to be a gas-protein reaction in the receptors in the synapses [4].”